# Permanent ferroelectric retention of BiFeO$_3$ mesocrystal

Ying-Hui Hsieh[1], Fei Xue[2], Tiannan Yang[2], Heng-Jui Liu[1], Yuanmin Zhu[3,4], Yi-Chun Chen[5], Qian Zhan[3], Chun-Gang Duan[6], Long-Qing Chen[2], Qing He[7] & Ying-Hao Chu[1,8]

Non-volatile electronic devices based on magnetoelectric multiferroics have triggered new possibilities of outperforming conventional devices for applications. However, ferroelectric reliability issues, such as imprint, retention and fatigue, must be solved before the realization of practical devices. In this study, everlasting ferroelectric retention in the heteroepitaxially constrained multiferroic mesocrystal is reported, suggesting a new approach to overcome the failure of ferroelectric retention. Studied by scanning probe microscopy and transmission electron microscopy, and supported via the phase-field simulations, the key to the success of ferroelectric retention is to prevent the crystal from ferroelastic deformation during the relaxation of the spontaneous polarization in a ferroelectric nanocrystal.

[1] Department of Materials Science and Engineering, National Chiao Tung University, Hsinchu 30010, Taiwan. [2] Department of Materials and Engineering, Pennsylvania State University, University Park, Pennsylvania 16802, USA. [3] School of Materials Science and Engineering, University of Science and Technology Beijing, Beijing 100083, China. [4] National Center for Electron Microscopy in Beijing, School of Materials Science and Engineering, Tsinghua University, Beijing 100084, China. [5] Department of Physics, National Cheng Kung University, Tainan 70101, Taiwan. [6] Key Lab of Polar Materials and Devices, Ministry of Education, East China Normal University, Shanghai 200241, China. [7] Department of Physics, Durham University, Durham DH1 3LE, UK. [8] Institute of Physics, Academia Sinica, Taipei 11529, Taiwan. Correspondence and requests for materials should be addressed to Y.-H.C. (email: yhc@nctu.edu.tw).

The success of achieving low-power consumption, multifunctional and green nanoelectronics relies on the advance in the electric-field control of lattice, charge, orbital and spin degrees of freedom[1]. A medium possessing the coupling between these degrees of freedom provides new solution to obtain more sophisticated control of these degrees of freedom. For instance, a variety of technology is developed based on the successful incorporation of ferroelectric and magnetic materials. To further enhance the performance or create new functionalities, electric-field control of ferromagnetism/spin forms an exciting new playground with a great potential to impact logic and memory devices, spintronics and high-frequency devices[2]. Recently, the use of magnetoelectric multiferroics, in which an electric field can be used to switch their magnetic order, deliver promising solutions and a rich spectrum of physics[3,4]. Multiferroics typically are insulators with an antiferromagnetic spin arrangement. Hence, the form of ferromagnet–multiferroic heterostructures has been investigated to achieve electric-field control of ferromagnetism[5]. Among the numerous multiferroic systems, both ferroelectric and antiferromagnetic orders with the spontaneous electrical polarization pointing along the pseudo-cubic $\langle 111 \rangle$ and the $G$-type antiferromagnetic spin configuration coexist in $BiFeO_3$ (BFO) at room temperature, making it appealing for the practical applications[6]. The orientation of the antiferromagnetic sub-lattice is always perpendicular to the direction of ferroelectric polarization of BFO, thus, there is a strong coupling couples between ferroelectric and antiferromagnetic orders[7]. A weak ferromagnetic moment can be induced due to the symmetry of BFO allowing a small canting of the Fe spins between the neighbour pseudo-cubic (111). The resulting magnetization is predicted to confine to the energetically degenerate {111} with the value of $\sim 0.05 \mu_B$ per unit cell[8]. A direct modulation of adjacent ferromagnetic layer in the multiferroic/ferromagnet heterostructure can be achieved through the combination of the intrinsic coupling between the ferroelectricity and antiferromagnetism and the coupling between the antiferromagnetism and corresponding weak ferromagnetism in BFO. The easy plane of magnetization can be modulated by switching the polarization through an electric field, thus, offering an exciting opportunity for controlling the spin through the application of an electric field[9].

Although BFO is a successful template for manipulating the spin degree of freedom via an electric field, before the realization of practical devices, several key issues are yet to be addressed. The primary control parameter in BFO is the ferroelectric order. Ferroelectric reliability issues, such as imprint, retention and fatigue have to be solved. For example, the retention is strongly correlated with the thermodynamic instability of ferroelectric states[10]. The formation of a depolarization field is induced when the polarization-induced bound charges are not fully screened. Thus, an unstable polarization state is formed due to the asymmetric free-energy landscapes between the polarizations pointing away and towards the substrate, typically attributed to the imbalance of the electrostatic boundary conditions. In the configuration of metal–ferroelectric–metal capacitors, the problem can be avoided by a careful design of the electrodes to ensure the balance of electrostatic boundaries. However, to reduce the size of the circuit, the structure of a ferroelectric/multiferroic transistor is more favourable in modern design of integrated circuit. In such a structure, one side of ferroelectric/multiferroic layer is typically in contact with a semiconductor; while the other side usually connects to a metal. The imbalance of the electrostatic boundary conditions causes a severe retention problem. To solve this problem, an additional energy term has to be incorporated into the system to balance the free-energy landscape[10]. Although the progress to

reduce the energy difference of the polarization double-well have been shown on related studies[11–13], ferroelectric retention is still a key issue yet to be overcome. In previous studies, attempts have been made to extend the ferroelectric retention of BFO[14,15]. For example, an enhancement of the retention has been discovered in the mixed-phase BFO films[15]. The phase boundaries act as pinning centres of domain walls during the relaxation process by taking the advantage of periodic energy potential, suggesting a possible route of using elastic energy to solve the ferroelectric retention problem.

Thus, in this study, we have designed a model system, a self-assembled BFO mesocrystal embedded in a $CoFe_2O_4$ (CFO) matrix, to utilize the elastic energy as a key parameter to solve the ferroelectric retention problem of BFO[16,17]. The intimate contact between the mesocrystal and matrix material provides a strong structural coupling[18,19]. This elastic energy term can be exploited to improve the ferroelectric retention since the ferroelectric switching of BFO typically involves an elastic deformation[20]. The achievement of an improvement in retention to a great extent in BFO can open up a new avenue for ferroelectric retention studies and the possible applications in electric-field controllable spintronic memory and logic devices.

## Results

### The idea of how to achieve permanent ferroelectric retention.
Two crucial requirements have to be fulfilled to achieve switchable, non-volatile magnetoelectric devices with BFO: (i) a selective control of the ferroelectric switching path; and (ii) the stability of the switched domains. The first requirement arises because each switching path in multiferroics may trigger different changes in the magnetic order, and some switching paths may not affect the magnetic order at all. In the case of BFO, only ferroelastic switching mediates the magnetoelectric coupling. It is very important to note that the 180° ferroelectric switching process in BFO is very different from a traditional one. In a traditional 180° switching process, the polarization reverses when the central atom, such as Zr/Ti in $Pb(Zr_{0.2}Ti_{0.8})O_3$ (PZT)[21,22], in a unit cell only moves along the direction of applied electric field passing through the central symmetric position. Such a relative displacement of the Zr/Ti ions from the centrosymmetric positions results a reversal of ferroelectric polarization as illustrated in Fig. 1a. On the contrary, based on the studies of ferroelectric switching in BFO, there is always an intermediate step involving a ferroelastic deformation during the 180° switching event regardless of BFO orientations[14,23,24]. The 180° switching process in BFO(111)[14,20,25] is schematically presented in Fig. 1b. During the application of a reversed bias, the cations move not only along the field direction but also sideways resulting in a rotation of the polarization vector besides the change of magnitude. Such a displacement of the cation towards sideways allows the polarization to pass through an intermediate state (central panel in Fig. 1b) during the switching process. In other words, the 180° switching process of BFO(111) is completed through a 71° switching event followed by a 109° switching event, a combination of rotation and magnitude change of the ferroelectric polarization along the out-of-plane (OOP) direction. Both the 71° and 109° switching events are accompanied by the ferroelastic deformation that mediate the magnetoelectric coupling. The switching processes in BFO can be characterized by polarization–electric field measurement with variable frequencies. An intermediate state (marked by the grey line in Fig. 1c) in BFO(111) can be observed during the polarization switching. The switchable ferroelectric polarization to this state is about $80 \mu C \, cm^{-2}$, which agrees with the two-step switching (a 71° switching followed by a 109° switching) process

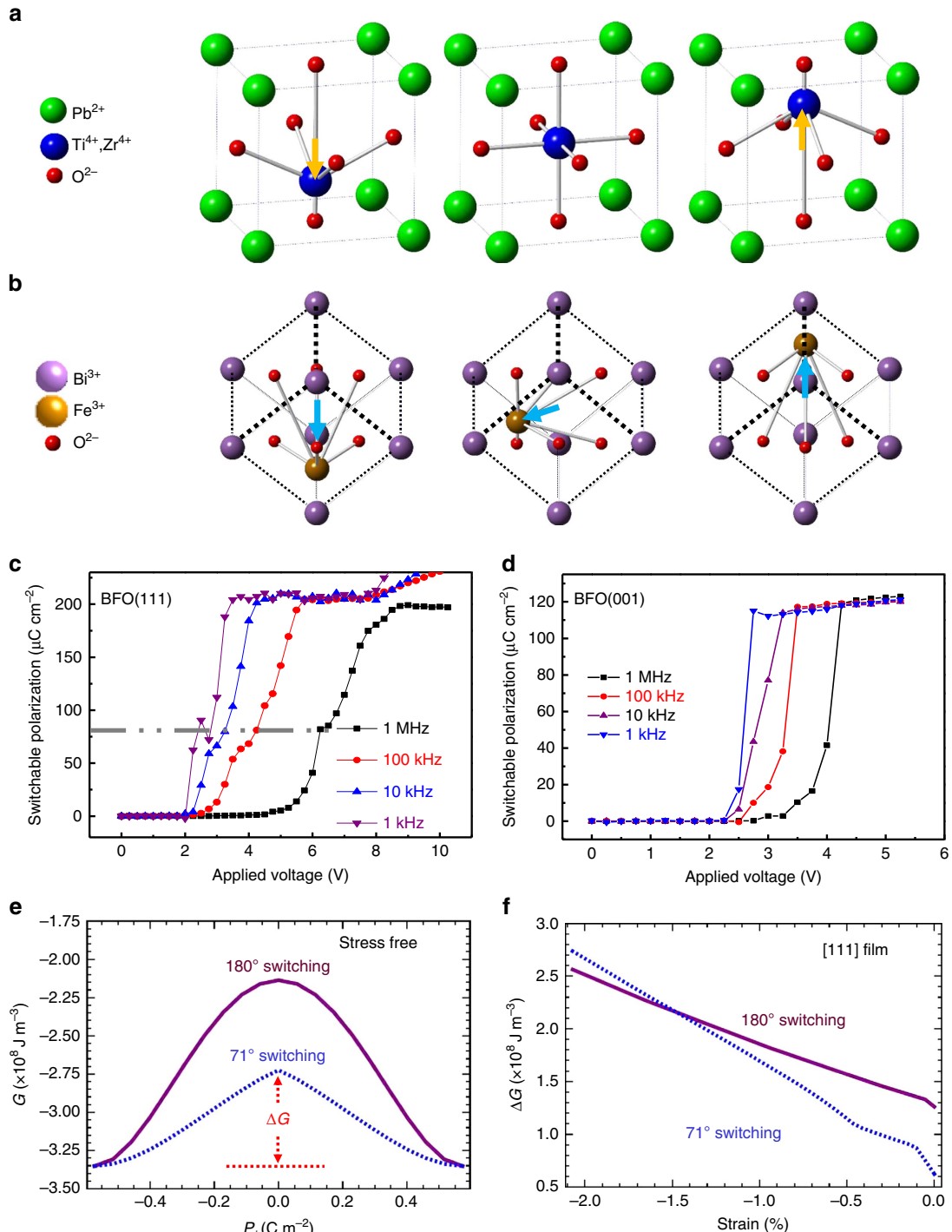

**Figure 1 | The processes of polarization switching and simulation of the effect of strain to polarization switching.** Schematic diagrams of polarization switching process for (**a**) PZT and (**b**) BFO(111), and the polarization–electric field PE loops for (**c**) BFO(111) and (**d**) BFO(001). (**e**) Total free energy as a function of switching path under stress-free boundary conditions. In 71° switching, the polarization direction is changed from [111] to [11$\bar{1}$] through [110], whereas in 180° switching, the polarization direction is switched from [111] to [$\bar{1}\bar{1}\bar{1}$] through [000]. (**f**) Energy barrier as a function of epitaxial strain in (111) BFO films. The arrows in (**a,b**) represent the direction of displacement of the central atoms ($Ti^{4+}$, $Zr^{4+}$ in PZT and $Fe^{3+}$ in BFO).

along the rhombohedral [111]. However, no intermediate state has been measured in the polarization–electric field measurement along [001] as showed in Fig. 1d since polarization–electric field measurement is only sensitive to the OOP component. Understanding the polarization switching process in BFO is very crucial because the accompanied ferroelastic deformation of the crystal is the key to stabilize the ferroelectric domain of BFO.

The second crucial requirement is the stability of the switched BFO domains. Local ferroelastic switching in BFO thin films produces a high-energy domain state compared with the surrounding domain[14], leading to a relaxation of the switched domain back to its original state. The relaxation is caused by the internal depolarization field without external stimulus[26]. This would impede long-term storage of information because the

magnetic order relaxes with the ferroelectric polarization or can even be decoupled. So far, the observed ferroelectric retentions of BFO thin films and the isolated BFO islands last only from few hours to about 1 day (ref. 27). Since the 180° switching of BFO(111) always accompanies the ferroelastic deformation, the prerequisite for the polarization relaxation is to conquer the elastic energy penalty originating from such a deformation. Therefore, a permanent retention can be realized only if one can prevent the ferroelastic deformation. In this study, the retention problem has been tackled effectively with the BFO(111) mesocrystal, an array of BFO nanocrystals with same crystal orientation embedded in a harder matrix. In this system, the stiff matrix is used for clamping down the deformation of the BFO crystal, which, in turn, suppresses the relaxation process. Therefore, the switching path can be controlled and the switched domain can be stabilized while each of BFO(111) nanocrystal is designed to be small enough to be fully switched without the existence of high-energy walls between the switched and surrounding domains.

To provide a theoretical support for the polarization stability of BFO(111) mesocrystal, we calculated the energy barrier that the system needs to cross during the switching process based on the thermodynamic analysis (for details, see Methods). As shown in Fig. 1e, for bulk BFO, the saddle point is [110] for the 71° switching and [000] for the trivial 180° switching, and the energy barrier is defined as the energy difference between the values at saddle point and the minimum point. The energy barrier of the 71° switching is much lower than that of the 180° switching even though the 71° switching involves ferroelastic deformation, the direct 180° switching is less favourable due to a large ion displacement[20], resulting in a two-step process of the 180° switching in BFO (Fig. 1c), in an agreement with the experiment. Particularly, the energy barriers of both 71° and 180° switching increase with an in-plane compressive strain. The 180° switching becomes more energetically favourable than the 71° switching for a large compressive strain for a BFO(111) film as shown in Fig. 1f because the in-plane compressive strain disfavours the polarization perpendicular to the pseudo-cubic [111], leading to an increase of the free energy at the saddle point of the 71° switching that is larger than that of at the saddle point of the 180° switching. On the basis of these results, the stability of BFO ferroelectric domains should be very sensitive to the elastic strain.

**Sample preparation and the investigation**. Triggered by the theoretical support, the BFO mesocrystal[28] with perfect crystallographic alignment along (111) is fabricated and embedded into a harder matrix to impose an additional stress on BFO nanocrystals for improving the retention. This kind of crystal-orientation-ordered superstructures has aroused wide interest due to their unique properties. Recent studies have focused on the self-assembled heteroepitaxial nanocomposites, wherein two immiscible oxide materials serve as nanopillars and matrix, respectively. The spontaneously assembled nanopillars are ordered in a perfect crystallographic orientation, which can be viewed as a two-dimensional mesocrystal. In these systems, the oxide mesocrystal and matrix can establish compact connection with each other, which can strongly affect the structural and physical properties of the mesocrystal[29]. Thus, the selection of the matrix materials is very important as they provide the flexibility of tailoring the crystal orientations[30] and physical properties of the mesocrystal. In this study, we demonstrated that in a self-assembled BFO–CFO mesocrystal, the constraints imposed by CFO matrix can provide an additional elastic energy term to stabilize the switched BFO domains since CFO is mechanically harder than BFO[30].

BFO(111) mesocrystal with a molar ratio of BFO:CFO = 2:1 was prepared on a SrRuO$_3$-buffered or 0.7%wt Nb-doped STO(111) substrate by pulsed laser deposition[31]. The growth temperature and pressure were set at 650 °C and 100 mTorr of pure oxygen, respectively. The energy density of laser was operated at 1 J cm$^{-2}$ with the repetition rate of 10 Hz. The schematic in Fig. 2a illustrates the typical structure of BFO mesocrystal grown on a Nb-doped STO(111) substrate. According to the Winterbottom reconstruction[32], the surface energy difference between CFO(111) and STO(111) is lower than that between BFO(111) and STO(111); therefore, CFO prefers to form the matrix and BFO becomes the mesocrystal[33–36]. Meanwhile, BFO remains tetrahedron on the surface due to its cubic equilibrium shape along (111), thus, leading to pyramid-shaped nanostructure on the surface as shown in the inset of Fig. 2b. The detailed microstructure examined by the cross-sectional transmission electron microscopy (TEM; Fig. 2b) shows a sharp interface between well-separated BFO nanocrystals and the CFO matrix. Detailed crystal structural analysis was then carried out by X-ray diffraction. The $\theta$–$2\theta$ scan along the surface normal (Fig. 2c) only shows the peaks of BFO(111) and CFO(222) in the vicinity of the STO(111) substrate, clearly indicating the spontaneous phase separation of BFO and CFO and well-aligned BFO(111) mesocrystal along the OOP direction. The strain state extracted from X-ray diffraction suggests that the BFO(111) mesocrystal suffers a tensile strain about 1% along the OOP direction (compressed in the in-plane direction), while the CFO matrix suffers a compressive strain about 0.3–0.4%. Combining the results of TEM and X-ray diffraction, we confirm that the BFO mesocrystal is constrained by the CFO matrix.

To confirm the ferroelectricity of the BFO(111) mesocrystal, piezoresponse force microscopy (PFM)[37] is an ideal approach for both probing and switching the local ferroelectric polarization at nanoscale. The box-in-box switched patterns were written on a set of BFO mesocrystal samples (as shown in Supplementary Figure 1), with the application of different electric fields via a conducting tip. In Fig. 2d, the red-boxed area was first poled with a −10 V tip bias, and followed with another poling with a +10 V tip bias in the green boxed area. Both the topography and the corresponding OOP polarization signals were continuously recorded over 20 months after the first poling (the images after 4,400 h are shown in Fig. 2d). The dark contrast indicates polarization pointing downward while the bright contrast means polarization in the upward direction. After the initial switching, we saw only dark contrast on the BFO nanocrystals outside the red box indicating a downward polarization of the mesocrystal in the as-grown state; the area between the red and green boxes shows bright contrast. These results suggest that all BFO nanocrystals are switchable, a confirmation of ferroelectricity in the BFO mesocrystal. To check the intermediate state during the 180° switching in BFO(111) mesocrystal, we have conducted PFM to switch the ferroelectricity of BFO nanocrystals as a function of the voltage pulse duration. The OOP and in-plane (IP) phase images before and after the application of voltage pulse are shown in Supplementary Note 1 and Supplementary Fig. 2. For the analysis of polarization direction and switching modes, the OOP and IP phase images in Supplementary Fig. 2a are combined and the results are presented as the schematics shown in Supplementary Fig. 2b. Here we compared the change of polarization and found the existence the 71° and 109° switching events, delivering a crucial evidence on the intermediate steps during the switching on this BFO system.

The coercive field and symmetry of ferroelectric hysteresis loop are key features to reflect the stability of the ferroelectric

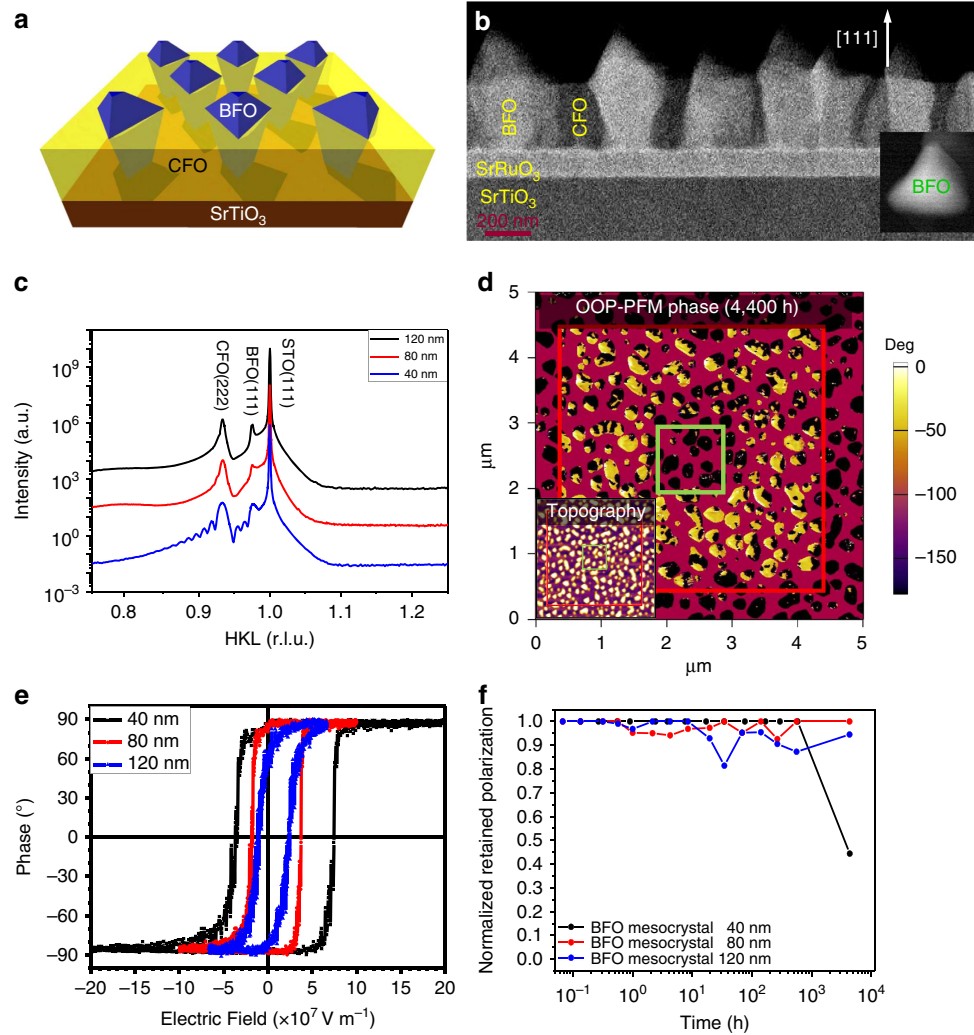

**Figure 2 | The fabrication/design of BFO(111) mesocrystal structure and the observation of retention behaviour. (a)** The schematic of BFO(111) mesocrystal. **(b)** The plane view of single BFO(111) nanocrystal and the cross-sectional TEM view of BFO(111) mesocrystals embedded in a CFO matrix. **(c)** High-resolution X-ray diffraction $\theta$–$2\theta$ scans around the STO(111) series of peaks. **(d)** The images of topology and corresponding OOP PFM recorded after switching 4,400 h, respectively. **(e)** Phase–voltage hysteresis loops of different sample thickness. **(f)** Comparison of normalized retained polarization versus the relaxing time among reversed domains switched in BFO mesocrystal of different thickness.

polarization. Thus, the phase–voltage hysteresis loops were measured through a Asylum Research Cypher S (ref. 38) system using Olympus PtIr-coated Si probes (spring constant $\sim 2\,\mathrm{N\,m}^{-1}$, tip radius $\sim 25\,\mathrm{nm}$) superimposed with a 1 Hz triangular square-stepping wave with bias window up to 8 V, and each loop was averaged over five consecutive cycles. For an ideal ferroelectric material with identical electrodes, the ferroelectric hysteresis loop should be symmetric with respect to the origin, meaning the same coercivity. However, in reality, the hysteresis loops measured from thin film or single crystal are usually asymmetric owing to the different electrostatic boundary conditions. The change in coercive field with various size and thickness of BFO nanocrystals has been compared (as shown in the Supplementary Note 2 and Supplementary Fig. 3). We found that at the same thickness, the shift of the coercive field is the same despite the size variation of BFO nanocrystals. As shown in Fig. 2e, the hysteresis loops of the BFO nanocrystals with different thickness are all non-centrosymmetric, suggesting a preference in the polarization direction. It is worth to note that the thickness of BFO mesocrystal can affect the deviation of the hysteresis loop. Most of the 40 nm-thick BFO nanocrystals have an obvious shift in the hysteresis loop from the origin (as shown in Supplementary

Fig. 3). Such a deviation is considered as a driving force in the ferroelectric relaxation since the shift of coercive fields suggests an existence of the depolarization field. When the polarization of BFO mesocrystal is switched, the depolarization field tends to reverse the polarization back to the original state. The effect of the depolarization field on the ferroelectric relaxation for different thicknesses can be observed in the Fig. 2f. The BFO mesocrystal of 40 nm thick relaxes much faster than the samples of 80 and 120 nm thick although the X-ray diffraction results showed that all these BFO mesocrystals suffer the same degree of strain. As mentioned earlier, the stress originated from the CFO matrix can prevent the reversal of polarization by clamping the structure of BFO nanocrystal. For completing this relaxation process that associates to the deformation of BFO crystal, it needs to conquer the stress from the CFO matrix. Hence, the BFO mesocrystal of 40 nm thick is easier to switch back since the depolarization bias is larger in a thinner sample to reserve the polarization to the original state. It should be noted that no relaxation was recorded on the 80 nm-thick BFO mesocrystal, suggesting a possible achievement of permanent ferroelectric retention.

To investigate the ferroelectric relaxation process of BFO mesocrystals, the evolution of the switched polarization was

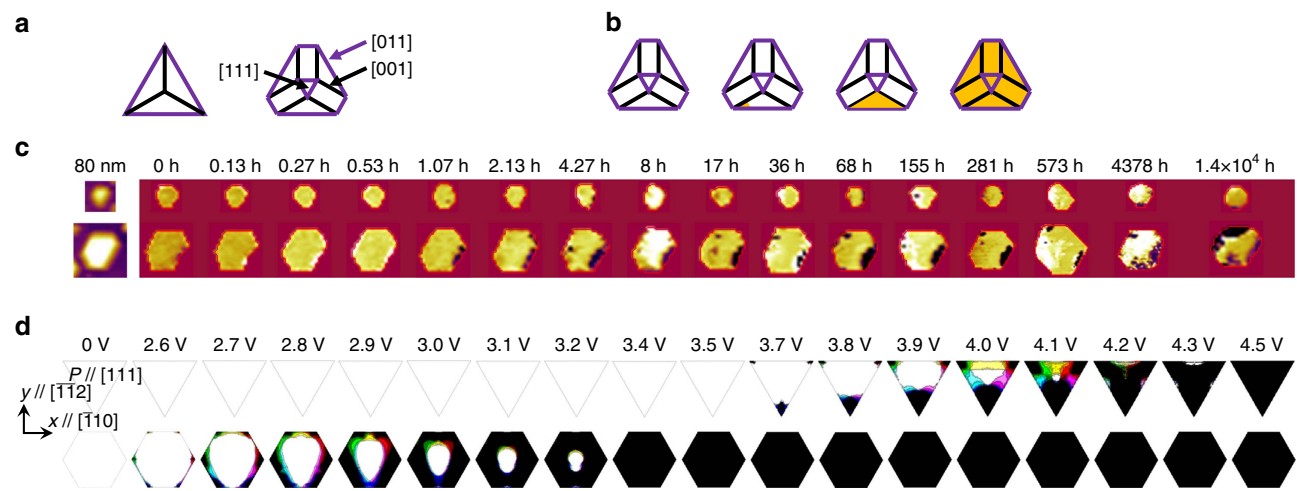

**Figure 3 | Different relaxation processes.** The schematics of (**a**) different shapes of BFO mesocrystal and (**b**) the evolution of relaxation process. (**c**) The relaxation process of BFO mesocrystal of different sizes as a function of time. (**d**) Switching process of different BFO mesocrystal shapes from the phase-field simulations, where white and black regions are ferroelectric domains with upward and downward polarization, respectively. Lateral area of the triangle and hexagonal mesocrystals are 0.01 and 0.04 $\mu m^2$, respectively. In the simulations, the electrostatic potential increases step by step, and the results are compared with experiments after different times.

recorded at specific time intervals (shown in Fig. 3c). As shown in the topography of the BFO mesocrystal (Fig. 2d), the BFO nanocrystals are not all identical in size, two most representative shapes/sizes (the area size of the big ones is $\sim 0.05\,\mu m^2$ and that of the small ones is $\sim 0.02\,\mu m^2$) could be found in most mesocrystals. The relaxation process of the most representative nanocrystals corresponding to the schematic in Fig. 3a from the sample of 80 nm-thick mesocrystal is shown in Fig. 3c. The up-row images in Fig. 3c present the relaxation process of the BFO nanocrystal with a smaller size and the triangular pyramid built with three (100) faces (the left panel in Fig. 3a). The bottom-row images in Fig. 3c show the relaxation process of the hexagonal BFO nanocrystal with larger size and a (111) facet on the top and three (110) faces (the right panel in Fig. 3a). From our observation, the BFO nanocrystals with smaller size maintain everlasting ferroelectric retention over the period of 1.5 year, while the others with larger size possessed poor ferroelectric retention and started to relax after about 1 h. The schematics of Fig. 3b illustrate the relaxation process of the BFO(111) nanocrystal. The relaxation process usually starts from the corner of the BFO nanocrystal (the yellow area in the second panel from left in Fig. 3b represents the reversal area). Once the switched polarization of the nanocrystals starts to relax, the relaxation process causes a propagation of ferroelectric domain walls leading to a retention failure. Moreover, the reversal area in Fig. 3c was further analysed through the IP and OOP PFM images (detailed discussion is provided in Supplementary Note 3, and Supplementary Figs 4 and 5) and the results show that the area with the 71° and 109° domain walls has a large chance to accelerate the reversal of switched polarization. This could be attributed to the larger elastic energy in the vicinity of these ferroelastic domain walls. Therefore, the appearance of these domain walls cannot stabilize the polarization retention, consistent with the previous study on ferroelectric fatigue issue of BFO films[25].

On the basis these results, the effect of the size/shape of BFO nanocrystals suggests that the larger BFO nanocrystals relax much faster than the smaller BFO nanocrystals regardless of the thickness. Even though the pyramid-shaped protrusions on top of the mesocrystal have no lateral constrains, which results in a possibility of mediating back-switching via the nucleation of opposite domains by these protrusions, the lateral constraints

in the bottom of BFO mesocrystal can still prevent the growth of opposite nucleated domains since an elastic deformation still needs to be involved to reverse the ferroelectric polarization in the constrained BFO. Moreover, the relative portion of the protrusion is small, therefore, even the switched polarization can be back-switched in the protrusion, the domain is not stable. Meanwhile, the switching dynamics predicted based on the phase-field simulations[39] agree very well with the experiments (for details of the simulation settings, see Methods). As shown in Fig. 3d, for the nanocrystals with both triangle and hexagonal shapes, the domains reverse by first nucleating at the boundaries of the BFO nanocrystals due to a reduced interfacial energy penalty, and then propagate inward till a full polarization reversal of the whole nanocrystal occurs. Furthermore, the switching field of the triangular nanocrystals is larger compared with that of the hexagonal nanocrystals, suggesting a superior ferroelectric stability of the triangular nanocrystals.

## Discussion
Figure 4a summarizes a comparison of the retention behaviours of BFO nanocrystals with different shapes/sizes, showing that the retention of BFO nanocrystal of small size is much longer than that of the large one. Figure 4b demonstrates the change of polarization phase as a function of electric field, and both the experiments and simulations show that the triangular nanocrystals have larger switching fields and better retention properties than those of the larger hexagonal nanocrystals. To investigate the origin of the different retention properties discovered in the triangular and hexagonal nanocrystals, we performed the phase-field simulations for BFO nanocrystals with different lateral areas. As seen in Fig. 4c, the coercive field increases with decreasing lateral area, while the shape of the nanocrystals makes negligible impact, indicating that the lateral size of nanocrystals is the dominant factor. To further reveal the underlying mechanism, the OOP stress in BFO nanocrystal is calculated as a function of the lateral area. As shown in Fig. 4d, the OOP tensile stress increases with a decreasing lateral area due to the increasingly strong clamping by the CFO matrix. The calculation result in Fig. 4d is supported by TEM results as shown in Supplementary Note 4 and Supplementary Fig. 6. The lattice misfit between CFO and BFO phases increases with the lateral

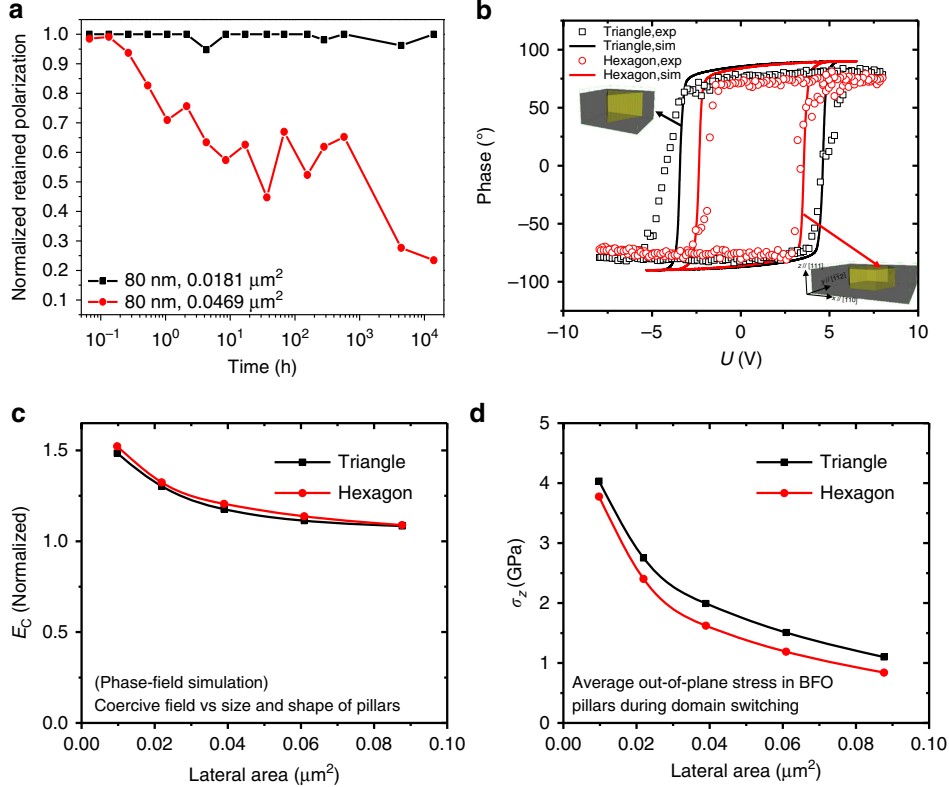

**Figure 4 | Polarization retention loss mechanism.** (**a**) Comparison of normalized retained polarization versus the relaxing time between BFO nanocrystals with different sizes. (**b**) Polarization phase changes as a function of electric fields. The open rectangles and circles represent experimental data, and the black and red lines are phase-field simulation results. (**c**) Coercive field as a function of the lateral area of the BFO mesocrystals with different shapes. The coercive field is normalized with respect to that of the BFO mesocrystals with the same thickness. (**d**) Average OOP stress as a function of the lateral area.

size of BFO nanocrystal. Besides, dislocations in the interface between CFO and large BFO nanocrystal were found, suggesting that the formation of dislocation mediates the strain relaxation, which cannot be found in the vicinity of small BFO nanocrystals. Therefore, nanocrystals with smaller area are subjected to larger in-plane compressive strain, which results in a stronger stabilization of the switched domains in BFO nanocrystals due to an increase of elastic energy, and gives rise to a larger energy barrier as shown in Fig. 1f. Furthermore, a comparison of ferroelectric retention behaviours among BFO mesocrystal and other ferroelectrics with asymmetric electrical boundary conditions presented in Supplementary Note 5 and Supplementary Fig. 7 shows that the BFO mesocrystal performs a predominant retention behaviour, confirming that the elastic energy can be the key to solve the retention failure.

To summarize, small BFO nanocrystals with the thickness of 80 nm exhibit the everlasting ferroelectric retention, revealing a much improved stability of the switched polarization state. The relaxation behaviours of other BFO nanocrystals of three different thicknesses were also characterized and analysed. The experimental results show that size and thickness of BFO nanocrystals affect the retention behaviour significantly. The effect of thickness shows a competition between the intrinsic depolarization field and the stress imposed by the CFO matrix, resulting in an optimal thickness of 80 nm. The BFO nanocrystals of smaller size suffer a strong clamping from the CFO matrix. Thus, the smaller BFO nanocrystals are much stable than the larger ones at the same thickness. The time constant of the still-remained BFO mesocrystal is infinite since it does not show any degradation, meaning the lifetime of the BFO mesocrystal is over 10 years of data retention. The prediction through the phase field simulation reveals that the ferroelastic deformation of BFO is

restricted by the stress originating from the intimate structural interaction between the BFO mesocrystal and CFO matrix. Therefore, the switched polarization in BFO nanocrystals cannot spontaneously reverse back. These results suggest that the approach of improving the ferroelectric retention by clamping the crystal structure is practical. The permanent ferroelectric retention of the strain-confined BFO mesocrystal presents a great leap towards realizing the non-volatile multiferroic devices.

## Methods
**Retention experiment.** To investigate the evolution of the retention behaviour in BFO mesocrystal, a set of negative ($-10$ V, $3 \times 3\,\mu m^2$)/positive ($+10$ V, $1 \times 1\,\mu m^2$) tip bias was used to switch the polarization of BFO mesocrystal (Supplementary Fig. 1), and then the polarization of BFO mesocrystal in the switched area was tracked by PFM with time. The duration of recording the polarization signal is twofold, such as 4, 8 and then 32 min. During recoding the PFM image, the information of topography would be recorded simultaneously. Therefore, we could overlap the location of BFO mesocrystal from the topography onto the PFM image. In this way, the evolution of domain structure of each BFO mesocrystal can be identified clearly as shown in Supplementary Fig. 1. The degree of relaxation of each single BFO mesocrystal is presented as the ratio of the area of remaining switched polarization to the area obtained from the corresponding topography. Then, recording the ratio of relaxation as a function of time for all the BFO mesocrystal, the evolution of domain structure with time and size can be plotted out as shown in Figs 2f and 4a.

**Phase-field simulation method.** The ferroelectric domain structure of BFO is described with a local polarization field $\mathbf{P}(\mathbf{r})$, where $\mathbf{r}$ is the position vector. Temporal evolution of the polarization field is described by the time-dependent Ginzburg–Landau equation, $\partial \mathbf{P}/\partial t = -L_P(\delta F/\delta \mathbf{P})$, which is solved numerically using the semi-implicit Fourier spectral method[40]. Where $L_P$ is a kinetic coefficient related to ferroelectric domain wall mobility[41]. $F = F_{Landau} + F_{gradient} + F_{elec} + F_{elastic}$ is the total free energy of the ferroelectric BFO. Here $F_{Landau}$ and $F_{gradient}$ are the ferroelectric Landau free energy and ferroelectric gradient energy, respectively, with their mathematical expressions given in refs 42,43. $F_{elec} = \int(-\frac{1}{2}\mathbf{E}^d \cdot \mathbf{P})d\mathbf{r}^3$ is the electrostatic energy, where $\mathbf{E}^d$ is the depolarization field simulated by solving the

electrostatic equation $\kappa_0 \kappa^b(\mathbf{r}) \nabla \cdot \mathbf{E}^d + \nabla \cdot \mathbf{P} = 0$ using a spectral iterative perturbation method[44]. $\kappa_0$ and $\kappa^b(\mathbf{r})$ denote the vacuum and background[45] (or in non-ferroelectric phases, relative) dielectric permittivity, respectively. An electrostatic boundary condition of given electric potential at top and bottom surfaces of the film is adopted to describe the applied electric field. The elastic energy $F_{\text{elastic}}$ is expressed as $F_{\text{elastic}} = \int \frac{1}{2} c_{ijkl}(\varepsilon_{ij} - \varepsilon_{ij}^0)(\varepsilon_{kl} - \varepsilon_{kl}^0) d\mathbf{r}^3$ (summation conventions over repeat indices ($i = 1,2,3$) are used), where $\mathbf{c}(\mathbf{r})$ is the local elastic stiffness tensor, $\varepsilon$ is the strain and $\varepsilon^0$ is the stress-free strain related to the local ferroelectric order $\varepsilon_{ij}^0 = Q_{ijkl}P_kP_l + \varepsilon_{ij}^{\text{lattice}}$, $\mathbf{Q}$ denoting the electrostrictive coefficient tensor. $\varepsilon(\mathbf{r})$ is simulated by solving the elastic equilibrium equation $(\partial / \partial r_j)(c_{ijkl}(\varepsilon_{kl} - \varepsilon_{kl}^0)) = 0$ under a thin film boundary condition, using a spectral iterative perturbation method[46] based on Khachaturyan's mesoscopic elasticity theory[47]. Material parameters (in SI units) in the crystal lattice coordinates are $\alpha_1 = 3.74 \times 10^8$, $\alpha_{11} = 2.29 \times 10^8$, $\alpha_{12} = 3.06 \times 10^8$, $\alpha_{111} = 5.99 \times 10^7$, $\alpha_{112} = -3.34 \times 10^5$, $\alpha_{123} = -1.78 \times 10^8$, $Q_{11} = 0.032$, $Q_{12} = -0.016$, $Q_{44} = 0.020$, $\kappa^b = 40$, $c_{11} = 228 \times 10^9$, $c_{12} = 128 \times 10^9$ for BFO, and $\kappa^b = 40$, $c_{11} = 286 \times 10^9$, $c_{12} = 173 \times 10^9$ for CFO.

**Thermodynamic calculations.** In the thermodynamic calculations, BFO is described by two sets of order parameters, that is, $P_i$ ($i = 1, 2, 3$) and $\theta_i$ ($i = 1, 2, 3$), which are the components of the spontaneous polarization and oxygen octahedral tilt in the pseudocubic coordinate system, respectively. The total free-energy density includes contributions from bulk free energy and elastic energy, and is given by[48,49]

$$f_{\text{single}} = \alpha_{ij}P_iP_j + \alpha_{ijkl}P_iP_jP_kP_l + \beta_{ij}\theta_i\theta_j + \beta_{ijkl}\theta_i\theta_j\theta_k\theta_l + t_{ijkl}P_iP_j\theta_k\theta_l \\ + \frac{1}{2}c_{ijkl}(\varepsilon_{ij} - \varepsilon_{ij}^0)(\varepsilon_{kl} - \varepsilon_{kl}^0), \tag{1}$$

where $\alpha_{ij}$, $\alpha_{ijkl}$, $\beta_{ij}$, $\beta_{ijkl}$ and $t_{ijkl}$ are coefficients of the Landau polynomial under the stress-free conditions, $c_{ijkl}$ is the elastic stiffness tensor and $\varepsilon_{ij}$ and $\varepsilon_{ij}^0$ are the total strain and eigen strain, respectively. The eigen strain is related to the polarization and oxygen octahedral tilt through $\varepsilon_{ij}^0 = \lambda_{ijkl}\theta_k\theta_l + h_{ijkl}P_kP_l$, where $\lambda_{ijkl}$ and $h_{ijkl}$ are coupling coefficients. All the coefficients are given in the Supplementary Materials of ref. 50.

Under stress-free conditions, the relation $\varepsilon_{ij} = \varepsilon_{ij}^0$ is maintained to eliminate the elastic energy contribution in equation (1). For constrained boundary conditions, the coordinate system is rotated according to Euler angles $[3\pi/4, \arccos[1/\sqrt{3}], 0]$, and in the new coordinate system, we use the fixed strain boundary conditions, $\varepsilon_{11} = \varepsilon_{22} = \varepsilon$, $\varepsilon_{33} = -6\varepsilon$, $\varepsilon_{12} = \varepsilon_{23} = \varepsilon_{13} = 0$ with $\varepsilon$ the biaxial strain.

**Data availability.** The data that support the findings of this study are available from the corresponding author on request.

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

## Acknowledgements

This work is supported by Ministry of Science and Technology, R.O.C. (104-2628-E-009-005-MY2 and 103-2119-M-009-003-MY3). The work at Penn State is supported by the NSF MRSEC under Grant No. DMR-1420620, and NSF under Grant No. DMR-1410714. The work in ECNU was supported by Ministry of Science and Technology of China (973 projects: Grant Nos. 2014CB921104 and 2013CB922301). The work at University of Science and Technology Beijing is supported by National Natural Science Foundation of China with Grant Nos. 51571021 and 51371031. The work at Durham University is supported by Engineering and Physical Sciences Research Council with Grant No. EP/N016718/1.

## Author contributions

Y.-H.H., Q.H., C.-G.D. and Y.-H.C. conceived and designed the experiments; Y.-H.H. carried out the sample preparation and SPM measurement; F.X., T.Y. and L.-Q.C. carried out the calculation; H.-J.L. performed the X-ray diffraction measurement; Q.Z. and Y.Z. contributed to the TEM experiment; Y.-H.H., F.X., T.Y., Y.-C.C., Q.H. and Y.-H.C. analysed the data; Y.-H.H., F.X., T.Y., Q.H. and Y.-H.C. co-wrote the paper.

## Additional information

**Competing financial interests:** The authors declare no competing financial interests.

