## [Peer Review File · Nature Communications]

Reviewers' comments:

Reviewer #1 (Remarks to the Author):

The authors suggest to rely on elastic constraints by a matrix surrounding BFO mesocrystals in order to address the challenge with poor polarization retention in multiferroic BFO. The working hypothesis is that the matrix prevents ferroelastic deformation, something that takes place during switching. Although an important problem to solve, and an original and relevant approach combining phase field simulations with thin film analysis is employed, if the paper is to be published in Nature Communications I suggest that the following points are addressed:

1. If elastic constraints governs the increased retention observed, what is the role of the pyramid shaped protrusions observed on top of the mesocrystals extending above the matrix? Those structures have no lateral constraints, and could thus mediate back-switching via nucleation of opposite domains.
2. In (111)-switching in BFO there are intermediate steps via 71 and 109 switching as the authors points out, and it has been shown that that can lead to fatigue. This also means that the in-plane polarization component is important. In the data shown, both in Fig. 2d and in Fig. 3c multiple levels of out of plane polarization seems to be detectable. Hence it is important that the authors address possible inclusions of 71 and 109 degree domains in their nanostructures via for example in-plane piezoelectric force microscopy studies. If such domains are present as the data seems to indicate, what is their role, and can they help to stabilize the polarization retention due to the surrounding matrix?
3. The approach proposed to increase retention is based on an elastic clamping effect via a hard matrix. The authors discuss the out-of-plane strain during switching as a function of lateral area of the mesocrystals, see for example Fig. 4d for phase field simulations. It would strengthen the paper if the author, for example via GPA analysis of TEM data, confirms experimentally the increased out of plane strain as the lateral size of the mesocrystals is decreased.

Reviewer #2 (Remarks to the Author):

This is a very interesting work on the retention of ferroelectric state of BFO mesocrystal through compressive stress, and will pave the way for practical memory devices. I recommend its publication, though there are number of points that need clarification:

1. On page 5, "Such a trajectory results a reversal of ferroelectricity and a deformation of the crystal in the field direction as illustrated in Fig. 1a." This is not entirely accurate, as the final state after switching, compared to initial one, has no macroscopic deformation.
2. The intermediate state in Fig. 1c is not entirely convincing. Is there other evidence supporting this claim? One possibility would be measuring strain versus field during the process.
3. On page 6, "The 180° switching of BFO(111) mesocrystal always accompanies the ferroelastic deformation." This is not entirely accurate as well. For example, in (001) BFO, as authors suggested, there would be no intermediate ferroelectric deformation.

4. It is difficult to understand "The energy barrier of the 71{degree sign} switching is much lower than that of the 180{degree sign} switching" As authors argued, 70 domain switching involves ferroelastic deformation, and thus the penalty should be higher. Why opposite? Any physical explanation?

5. The fact that 40nm thick system requires larger coercive field to switching seem to contradict the claim that it relaxes faster. Larger coercive field usually mean more stable system. Any clarification?

Reviewer #3 (Remarks to the Author):

This paper presents a novel and interesting study of the effect of crystal size and geometry and retention in self assembled BFO-CFO mesocrystalline systems.

Although the results are interesting, the authors dramatically oversell their significance. All ferroelectrics under appropriate boundary conditions should have permanent ferroelectric polarization. It is thus somewhat of a "straw man" argument to create a system where you have poor retention and then tune it until you have good retention. Further, while measuring a years retention is very good, I do not think the authors should describe it as "everlasting". This seems quite over-dramatic, "extremely long" would be sufficient

Another serious deficiency of the paper is what seems to be a willful neglect of the previous literature on BFO-CFO nanocomposite systems. There have been literally hundreds of papers on this since the 2005 Nanoletters paper of Zavaliche et al (Nanoletters, 5 1793 (2005)) from Ramesh's group (and of course the 2004 pioneering paper in Science from Zheng et al in 2004 on BTO-CFO, also from Ramesh's group). One especially relevant to this paper is Zheng et al, Advanced Materials 18,2747 (2006), where the fabrication of the 111 oriented structures studied here was described. It is inconceivable that the authors are not aware of this because the corresponding author of this work is on that paper! There are numerous investigations of these kinds of systems using PFM (unsurprising given that the chief interest is coupling between polarization and magnetism through strain in this system and the lead driver has been the Ramesh group in Berkeley who are experts in PFM), none of which are cited. One in particular stands out, Zheng et al, Nanoletters, 6, 1405 (2006), where the PFM response of BFO-CFO heterostructures on 111 substrates was measured. Curiously, when this was published 10 years ago, nothing was mentioned about the retention, either good or bad.

As the corresponding author was a postdoc in the Ramesh group and indeed co-authored many of the relevant works in the area, I can only conclude the large body of work on BFO-CFO nanostructures has been intentionally neglected to create an exaggerated sense of the novelty of this work.

My recommendation is that this work not be accepted for publication in Nature Communications. I would like to see the authors rewrite it without all the unnecessary spin, and include instead a sufficient discussion and citations to previous work. they should then submit it to a more specialized journal. If they do this, I do think the community will find the results to be of interest. I should note that in the process the authors would do well to seek a proofread from a native speaker of English as this paper is literally full of grammatical mistakes. As some of the authors have US and UK addresses, I am sure that this would not be too difficult to do and really should be done any time that a group of non-native speakers submits a paper for publication.

REVIEWERS' COMMENTS:

Reviewer #1 (Remarks to the Author):

After the revisions by the authors I can recommend this paper to be published.

Reviewer #2 (Remarks to the Author):

I have reviewed the revised manuscript, as well as the authors' response, and I am happy with the authors' revision. I would suggest that the authors include more references on PFM, phase field simulations, and BFO, to make it more representative of the field.

Revision requested for the manuscript No. NCOMMS-16-04021A

Response to Referee #1

Reviewer Comment:

The authors suggest to rely on elastic constrains by a matrix surrounding BFO mesocrystals in order to address the challenge with poor polarization retention in multiferroic BFO. The working hypothesis is that the matrix prevents ferroelastic deformation, something that takes place during switching. Although an important problem to solve, and an original and relevant approach combining phase field simulations with thin film analysis is employed, if the paper is to be published in Nature Communications I suggest that the following points are addressed.

Response:

We thank the referee for the support on this manuscript. The questions are addressed in the following response.

Reviewer Comment:

1. If elastic constraints govern the increased retention observed, what is the role of the pyramid shaped protrusions observed on top of the mesocrystals extending above the matrix? Those structures have no lateral constraints, and could thus mediate back-switching via nucleation of opposite domains.

Response:

We do agree with the reviewer that the pyramid shaped protrusions on top of the mesocrystal have no lateral constrains. Thus, there is a possibility of mediating back-switching via the nucleation of opposite domains by these protrusions. However, the lateral constraints in the bottom of BFO mesocrystal can still prevent the growth of opposite nucleated domains since an elastic deformation still needs to be involved to reverse the ferroelectric polarization in the constrained BFO. In addition, the relative portion of the protrusion is small, therefore, even they can be back-switched, the domains are not stable. We have included the corresponding discussion in the revised manuscript.

Reviewer Comment:

2. In (111)-switching in BFO there are intermediate steps via 71 and 109 switching as the authors points out, and it has been shown that they can lead to fatigue. This also

means that the in-plane polarization component is important. In the data shown, both in Fig. 2d and in Fig. 3c multiple levels of out of plane polarization seems to be detectable. Hence it is important that the authors address possible inclusions of 71 and 109 degree domains in their nanostructures via for example in-plane piezoelectric force microscopy studies. If such domains are present as the data seems to indicate, what is their role, and can they help to stabilize the polarization retention due to the surrounding matrix?

Response:

We thank the referee for reminding the importance of reliability issues caused by the 71° and 109° switching events. We have gone back to our old data to compare the contrast of OOP and IP PFM images. Indeed, there exist 71°, 109°, and non-neutral domain walls during the relaxation process as shown in **Fig. R1**.

Figure R1. The OOP and IP PFM images of BFO nanocrystal.

Therefore, additional experiments have been conducted. We traced the change of the polarization in BFO nanocrystals during the relaxation by PFM as shown in **Fig. R2**. After the switching, the contrast in OOP PFM is bright and the one in IP PFM is brown, meaning the polarization points upwards. Then the polarization starts to reverse back as time passes. During the relaxation, it shows a multi-step switching process since the contrast in both IP and OOP PFM images presents three colors. According to the comparison of the OOP and IP PFM images, we can figure out the types of domain walls. From the observation, the area with the 71° and 109° domain wall has a large chance to reverse the polarization back. This could be attributed to the larger elastic energy in the vicinity of these ferroelastic domain walls. Therefore, the appearance of these domain walls cannot help to stabilize the polarization retention. All these new results are included in the supporting materials. A short discussion on the role of 71 and 109 domain walls is also included in the revised manuscript.

Figure R2. The change of polarization of BFO nanocrystal during the relaxation. (a) The OOP and the corresponding (b) IP PFM images. (c) The distribution of domain walls in IP PFM images. Black, brown and grey dotted lines correspond to dark, purple and bright contrasts in the OP image, respectively. Sky-blue lines are the non-neutral domain wall.

Reviewer Comment:

3. The approach proposed to increase retention is based on an elastic clamping effect via a hard matrix. The authors discuss the out-of-plane strain during switching as a function of lateral area of the mesocrystals, see for example Fig. 4d for phase field simulations. It would strengthen the paper if the author, for example via GPA analysis of TEM data, confirms experimentally the increased out of plane strain as the lateral size of the mesocrystals is decreased.

Response:

We thank the reviewer for this relevant question. Actually, the out-of-plane strain is highly coupled with the in-plane one, since BFO nanocrystal is treated as an elastic object with the strain imposed by the CFO matrix. Thus, it is more important to compare the in-plane strain state in the as-grown state. In order to realize this, we have carried out additional TEM and Geometric Phase Analysis (GPA)¹ analysis on the BFO nanocrystals with different sizes. **Fig. R3** (a) and **R3** (f) are the plane-view HRTEM images of a 120 nm-thick BFO mesocrystal. The GPA¹ method, which is an image processing technology used for mapping lattice displacement and has been successfully applied to characterize interface or defect structures such as misfit dislocations and their associated strain fields, was carried out based on the HRTEM images at the heterogeneous interfaces. The strain maps of the BFO-CFO heterointerface shown in **Fig. R3** (b), (c), (g), and (h) were performed at the interface between CFO matrix and BFO nanocrystals of diameter ~15 nm and ~170 nm, respectively. For the e_{xx} , e_{yy} distortion maps ($\langle 110 \rangle$, $\langle 1-12 \rangle$ directions) of small BFO

nanocrystal (~ 15 nm) as shown in **Fig. R3** (b) and **R3** (c), the uniform color contrast was revealed with a lattice misfit of $\sim 2.0\%$ in e_{xx} (**Fig. R3** (d)) and $\sim 2.5\%$ in e_{yy} (**Fig. R3** (d)) between the CFO and BFO phases. This suggests that the BFO nanocrystal is constrained by a strong strain from the matrix. While for the distortion maps (e_{xx} , e_{yy}) of large BFO nanocrystal (~ 170 nm) as shown in **Figs. R3** (g) and **R3** (h), the obvious color contrast was revealed with a lattice misfit of $\sim 6.0\%$ in e_{xx} and e_{yy} maps (**Fig. R3** (i)) between the CFO and BFO phases. The value of lattice misfit is close to the calculated misfit using their bulk values, revealing that the lattices of BFO nanocrystal are almost relaxed. In addition, dislocations can be found at the interface between CFO matrix and large BFO nanocrystals (**Fig. R3** (j)), suggesting the strain relaxation is mediated by the formation of dislocations, which cannot be found in the vicinity of small BFO nanocrystals (**Fig. R3** (e)). This explains why large BFO nanocrystals relax much faster than small ones. We have included these results in the support materials. The corresponding discussion has also been incorporated in the revised manuscript.

Figure R3. The TEM results of different size of BFO nanocrystal with (a-e) ~ 15 nm and (f-j) 170 nm in the sample of thickness 120nm, respectively. (a) and (f) The TEM images; (b), (c), (g) and (h) the corresponding strain mappings (e_{xx} and e_{yy}) of the TEM images in (a) and (f); (d) and (i) the corresponding line scan of the lattice misfit of the white lines in (b), (c), (g) and (h); (e) and (j) the Fourier filtered images.

Reference

1. Hýtcha, M. J., Snoeckb, E., Kilaasc, R. Quantitative measurement of displacement and strain fields from HREM micrographs. *Ultramicroscopy* **74**, 131-146 (1998).

Response to Referee #2

Reviewer Comment:

This is a very interesting work on the retention of ferroelectric state of BFO mesocrystal through compressive stress, and will pave the way for practical memory devices. I recommend its publication, though there are number of points that need clarification.

Response:

We sincerely appreciate the support on this manuscript. All the points have been carefully revisited and addressed in the following response.

Reviewer Comment:

1. On page 5, "Such a trajectory results a reversal of ferroelectricity and a deformation of the crystal in the field direction as illustrated in Fig. 1a." This is not entirely accurate, as the final state after switching, compared to initial one, has no macroscopic deformation.

Response:

Thank you very much for pointing out this. There is no macroscopic deformation after the ferroelectric switching. We are very sorry for the mistake and we have modified it to "Such a relative displacement of the Zr/Ti from the centrosymmetric positions results a reversal of ferroelectricity as illustrated in Fig. 1a."

Reviewer Comment:

2. The intermediate state in Fig. 1c is not entirely convincing. Is there other evidence supporting this claim? One possibility would be measuring strain versus field during the process.

Response:

We thank the reviewer for this suggestion. We have thought to measure the strain versus field during the switching process. However, the ferroelectric switching typically happens in a time scale less than 1 micro-second, therefore, it is very difficult to pick up this signal. In addition, due to the leakage problem, it is very hard to do a large area poling. More importantly, the retention behavior strongly depends on the size of BFO nanocrystal. Therefore, a technique with spatial resolution is required. In order to verify this, we have conducted PFM to switch the ferroelectricity

of BFO nanocrystals as a function of voltage pulse duration. The OOP and IP phase images before and after the application of voltage pulse are shown in **Fig. R4** (a). For the analysis of polarization direction and switching modes, the OOP and IP phase images in **Fig. R4** (a) are combined and the results are presented as the schematics shown in **Fig. R4** (b). Here, we compared the change of polarization and found the existence of the 71° and 109° switching events, delivering more evidence on the intermediate steps during the switching. However, if a large voltage with long pulse is applied, BFO nanocrystals are fully 180° switched, which is the initial state for our retention study. We have included these new data in the supporting materials. A short discussion is also included in the revised manuscript.

Figure R4. (a) The out-of-plane (OP) and in-plane (IP) phase images of BFO mesocrystal before and after hit by a voltage pulse. (b) The schematics of the combination of OP and IP phase images in (a). The arrows use to present the orientation of in-plane polarization.

Reviewer Comment:

3. On page 6, "The 180° switching of BFO(111) mesocrystal always accompanies the ferroelastic deformation." This is not entirely accurate as well. For example, in (001) BFO, as authors suggested, there would be no intermediate ferroelectric deformation.

Response:

We are sorry for this confusion. According to the early study by Kubel and Schmid on BFO single crystal¹, a direct 180° ferroelectric switching in BFO is less favorable due to a larger ion displacement compared to the other switching events. Based on the studies of ferroelectric switching in BFO, there is always an intermediate step involving a ferroelastic deformation during the 180° switching event regardless of BFO orientations²⁻⁴. In the measurement of the PE loops, it is only sensitive to the out-of-plane component, since a configuration of top and bottom electrodes is used. Therefore, even there is an intermediate step in BFO(001) film, we cannot pick it up. We have modified the corresponding text to clarify this issue.

References

1. Kubel, F, and Schmid, H. Structure of a Ferroelectric and Ferroelastic Monodomain Crystal of the Perovskite BiFeO₃. *Acta Crystallogr., Sect. B: Struct. Sci.* **46**, 698 (1990)
2. Cruz, M. P. *et al.* Strain Control of Domain-Wall Stability in Epitaxial BiFeO₃ (110) Films. *Phys. Rev. Lett.* **99**, 217601 (2007).
3. Baek, S. H. *et al.* Ferroelastic switching for nanoscale non-volatile magnetoelectric devices. *Nature Mater.* **9**, 309–314 (2010).
4. Heron, J. T. *et al.* Deterministic switching of ferromagnetism at room temperature using an electric field. *Nature* **516**, 370–373 (2014).

Reviewer Comment:

4. It is difficult to understand "The energy barrier of the 71° switching is much lower than that of the 180° switching" As authors argued, 71 domain switching involves ferroelastic deformation, and thus the penalty should be higher. Why opposite? Any physical explanation?

Response:

Based on the early study by Kubel and Schmid on BFO single crystal¹, the 180° ferroelectric switching in BFO could be less favorable due to a larger ion displacement compared to the other switching events. From Table R1, the displacements of iron and oxygen ions are smaller for 60° and 120° switching (corresponding to the 71° and 109° switching respectively). Therefore, these new ion positions seem to be preferred compared to the ion positions for the 180° switching event. However, as the reviewer mentioned, it has been a debate on whether ferroelastic switching in BFO is more favorable since the energy penalty has to be paid. Later on, there are more reports providing the strong experimental evidence on

the intermediate switching step²⁻⁴, which is a crucial step to the reversal of ferroelectric polarization in BFO.

Switching	Fe	O		Bi	
		+ α	- α	+ α	- α
60°	0.44	0.87	0.89	0.04	0.04
120°	0.62	1.13	1.24	0.04	0.06
180°	0.82	1.34	1.53	0.0	0.0

Table R1. Ion displacements for different switching modes^[1].

References

1. Kubel, F, and Schmid, H. Structure of a Ferroelectric and Ferroelastic Monodomain Crystal of the Perovskite BiFeO₃. *Acta Crystallogr., Sect. B: Struct. Sci.* **46**, 698 (1990)
2. Cruz, M. P. *et al.* Strain Control of Domain-Wall Stability in Epitaxial BiFeO₃ (110) Films. *Phys. Rev. Lett.* **99**, 217601 (2007).
3. Baek, S. H. *et al.* Ferroelastic switching for nanoscale non-volatile magnetoelectric devices. *Nature Mater.* **9**, 309–314 (2010).
4. Heron, J. T. *et al.* Deterministic switching of ferromagnetism at room temperature using an electric field. *Nature* **516**, 370–373 (2014).

Reviewer Comment:

5. The fact that 40 nm thick system requires larger coercive field to switching seem to contradict the claim that it relaxes faster. Larger coercive field usually mean more stable system. Any clarification?

Response:

Although the 40 nm thick system does show a larger coercive field, the deviation of the center of the hysteresis loop from the origin is also larger compared to the other two systems as shown in **Fig. R5**. The deviation is considered as a driving force of the ferroelectric relaxation since it suggests an existence of the depolarization field.

Figure R5. The distribution of the deviation of the center of the hysteresis loop of the three thicknesses systems with different BiFeO₃ mesocrystal size. The center of the hysteresis loop is calculated as $[(+E_{\text{coercive field}}) + (-E_{\text{coercive field}})]/2$.

Response to Referee #3

Reviewer Comment:

This paper presents a novel and interesting study of the effect of crystal size and geometry and retention in self assembled BFO-CFO mesocrystalline systems.

Response:

We thank the referee for the positive response.

Reviewer Comment:

Although the results are interesting, the authors dramatically oversell their significance. All ferroelectrics under appropriate boundary conditions should have permanent ferroelectric polarization. It is thus somewhat of a "straw man" argument to create a system where you have poor retention and then tune it until you have good retention. Further, while measuring a year retention is very good, I do not think the authors should describe it as "everlasting". This seems quite over-dramatic, "extremely long" would be sufficient.

Response:

In principle, there is no ferroelectric retention problem if appropriate electrical boundary conditions can be implemented. In a typical metal-ferroelectric-metal capacitor, the problem can be solved when a structure of symmetric electrodes is used¹. However, there are several other configurations for practical applications, such as metal-ferroelectric-semiconductor and AFM tip/ferroelectric/metal, in which the symmetric electrodes cannot be made. Therefore, a severe retention problem occurs. We have tracked this problem on BFO for a while²⁻⁴ and a serious retention problem can be observed on BFO films with various orientations. **Fig. R6** shows the comparison on the ferroelectric retention of BFO films, our BFO mesocrystal, and other ferroelectric materials with asymmetric electrical boundaries, suggesting that the ferroelectric retention is a generic problem. A new mechanism should be incorporated to provide a solution to this problem. This sets the novelty of this work. We provide a possible solution to this long-term issue, which hinders the applications of ferroelectrics in the past and multiferroics in the future. The relaxation of ferroelectric polarization follows certain physical principles. Typically, the characteristic time of relaxation can be extracted based on the initial relaxation trend via various theoretical model. However, in our system, it doesn't show any

degradation. Thus, we can say the retention time is much longer than our measuring period, one order longer at least. We call it "permanent", because it can last longer than the regular device life. However, if the reviewer insists, we will modify it. We have also included **Fig. R6** in the supporting materials.

Figure R6. Comparison of normalized retained polarization versus the relaxing time between BFO mesocrystal and other ferroelectrics in earlier reports⁴⁻¹⁴.

References

1. Y. H. Chu, Q. Zhan, C.-H. Yang, M. P. Cruz, L. W. Martin, T. Zhao, P. Yu, R. Ramesh, P. T. Joseph, I. N. Lin, W. Tian, and D. G. Schlom. Low voltage performance of epitaxial BiFeO₃ films on Si substrates through lanthanum substitution, *Appl. Phys. Lett.* **92**, 102909 (2008)
2. Y.-C. Chen, Q.-R. Lin and Y.-H. Chu. Domain growth dynamics in single-domain-like thin films, *Appl. Phys. Lett.* **94**, 122908 (2009)
3. Y.-C. Chen, C.-H. Ko, Y.-C. Huang, J.-C. Yang and Y.-H. Chu. Domain relaxation dynamics in epitaxial BiFeO₃ films: Role of surface charges, *J. Appl. Phys.* **112**, 052017 (2012)
4. Y.-C. Huang, Y. Liu, Y.-T. Lin, H.-J. Liu, Q. He, J. Li, Y.-C. Chen, Y.-H. Chu. Giant Enhancement of Ferroelectric Retention in BiFeO₃ Mixed-Phase Boundary, *Adv. Mater.* **26**, 6335–6340 (2014)
5. T. K. Song, J. G. Yoon, S. I. Kwun. Microscopic Polarization Retention Properties of Ferroelectric Pb(Zr,Ti)O₃ Thin Films, *Ferroelectrics* **335**, 61-68 (2006)

6. D. S. Fu, K. Suzuki, K. Kato, H. Suzuki. Dynamics of nanoscale polarization backswitching in tetragonal lead zirconate titanate thin film, *Appl. Phys. Lett.* **82**, 2130 (2003).
7. Y. Kan, X. Lu, H. Bo, F. Huang, X. Wu, J. Zhu. Critical radii of ferroelectric domains for different decay processes in LiNbO₃ crystals, *Appl. Phys. Lett.* **91**, 132902 (2007).
8. J. W. Hong, W. Jo, D. C. Kim, S. M. Cho, H. J. Nam, H. M. Lee, J. U. Bu. Nanoscale investigation of domain retention in preferentially oriented PbZr_{0.53}Ti_{0.47}O₃ thin films on Pt and on LaNiO₃, *Appl. Phys. Lett.* **75**, 75, 3183 (1999).
9. C. S. Ganpule, V. Nagarajan, S. B. Ogale, A. L. Roytburd, E. D. Williams, R. Ramesh. Domain nucleation and relaxation kinetics in ferroelectric thin films, *Appl. Phys. Lett.* **77**, 3275 (2000).
10. A. Gruverman, H. Tokumoto, A. S. Prakash, S. Aggarwal, B. Yang, M. Wuttig, R. Ramesh, O. Auciello, T. Venkatesan. Nanoscale imaging of domain dynamics and retention in ferroelectric thin films, *Appl. Phys. Lett.* **71**, 3492 (1997).
11. H. R. Zeng, K. Shimamura, E. G. Villora, S. Takekawa, K. Kitamura. Domain growth kinetics and wall strain behavior in BaMgF₄ ferroelectric crystal by piezoresponse force microscopy, *J. Appl. Phys.* **101**, 074109 (2007).
12. A. Gruverman, M. Tanaka. Polarization retention in SrBi₂Ta₂O₉ thin films investigated at nanoscale, *J. Appl. Phys.* **89**, 1836 (2001).
13. C. S. Ganpule, A. L. Roytburd, V. Nagarajan, B. K. Hill, S. B. Ogale, E. D. Williams, R. Ramesh, J. F. Scott. Polarization relaxation kinetics and 180° domain wall dynamics in ferroelectric thin films, *Phys. Rev. B* **65**, 014101 (2001).
14. V. V. Shvartsman, A. L. Kholkin, M. Tyunina, J. Levoska, Relaxation of induced polar state in relaxor PbMg_{1/3}Nb_{2/3}O₃ thin films studied by piezoresponse force microscopy, *Appl. Phys. Lett.* **86**, 222907 (2005).

Reviewer Comment:

Another serious deficiency of the paper is what seems to be a willful neglect of the previous literature on BFO-CFO nanocomposite systems. There have been literally hundreds of papers on this since the 2005 Nanoletters paper of Zavaliche et al (Nanoletters, 5 1793 (2005)) from Ramesh's group (and of course the 2004 pioneering paper in Science from Zheng et al in 2004 on BTO-CFO, also from Ramesh's group). One especially relevant to this paper is Zheng et al, *Advanced Materials* 18,2747 (2006), where the fabrication of the 111 oriented structures studied here was described. It is inconceivable that the authors are not aware of this because the corresponding author of this work is on that paper! There are numerous investigations of these kinds of systems using PFM (unsurprising given that the chief interest is

coupling between polarization and magnetism through strain in this system and the lead driver has been the Ramesh group in Berkeley who are experts in PFM), none of which are cited. One in particular stands out, Zheng et al, Nanoletters, 6, 1405 (2006), where the PFM response of BFO-CFO heterostructures on 111 substrates was measured. Curiously, when this was published 10 years ago, nothing was mentioned about the retention, either good or bad.

Response:

As the reviewer said, I am in this field for a while. I am aware of all the publications on the similar systems. In the early studies, the focus has been paid on the growth control and the magnetoelectric coupling on this system. Most of the studies were conducted on the BFO-CFO system on STO(001), in which BFO is the matrix, CFO forms nanopillars opposite to our case. Although PFM has been shown on this system, there is no study addressing the ferroelectric retention issue, which remains as a critical issue to practical applications. I will be happy to withdraw this paper if the reviewer can point out a study on the same issue with the similar system.

Reviewer Comment:

As the corresponding author was a postdoc in the Ramesh group and indeed co-authored many of the relevant works in the area, I can only conclude the large body of work on BFO-CFO nanostructures has been intentionally neglected to create an exaggerated sense of the novelty of this work.

Response:

I am very sorry and surprised that the reviewer has such an impression on my study. The major reason that I did not cite these papers is that they are not very relevant to this study (not related to the issue of ferroelectric retention). Moreover, based on my training, we always avoid the self-citation to increase our own citation numbers. So, definitely there is no such an intention to ignore previous work to create an exaggerated sense of the novelty of this work. In order to fix this issue, some relevant references related to the growth part have been included in the revised manuscript. I believe the novelty of this study has been addressed very carefully in the introduction part. We suggested a possible solution of ferroelectric retention problem, which is a very critical issue to solve if one wants to use BFO in practical applications.

Reviewer Comment:

My recommendation is that this work not be accepted for publication in Nature Communications. I would like to see the authors rewrite it without all the unnecessary spin, and include instead a sufficient discussion and citations to previous work. they should then submit it to a more specialized journal. If they do this, I do think the community will find the results to be of interest. I should note that in the process the authors would do well to seek a proofread from a native speaker of English as this paper is literally full of grammatical mistakes. As some of the authors have US and UK addresses, I am sure that this would not be too difficult to do and really should be done any time that a group of non-native speakers submits a paper for publication.

Response:

We have revised the manuscript and included some previous studies in the references. The proof reading and clarity in language have been improved. We hope with the improvement the reviewer can recommend the publication of this manuscript in Nature Communications.